# The Influence of the Long-Term Outdoor Storage of Rockrose (*Cistus laurifolius* L.) Shrub Biomass on Biofuel’s Quality, Pre-Treatment and Combustion Processes

**DOI:** 10.3390/biology12111451

**Published:** 2023-11-19

**Authors:** Raquel Bados, Irene Mediavilla, Eduardo Tolosana, Elena Borjabad, Raquel Ramos, Miguel José Fernández, Paloma Pérez, Luis Saúl Esteban

**Affiliations:** 1CEDER-CIEMAT, Centro de Desarrollo de Energías Renovables—Centro de Investigaciones Energéticas Medioambientales y Tecnológicas, Autovía A-15, Salida 56, 42290 Lubia, Spain; irene.mediavilla@ciemat.es (I.M.); elena.borjabad@ciemat.es (E.B.); raquel.ramos@ciemat.es (R.R.); miguel.fernandez@ciemat.es (M.J.F.); paloma.perez@ciemat.es (P.P.); luis.esteban@ciemat.es (L.S.E.); 2Departamento de Ingeniería y Gestión Forestal y Ambiental, E.T.S.I. Montes, Forestal y del Medio Natural, Universidad Politécnica De Madrid, C/José Antonio Nováis 10, Campus Ciudad Universitaria, 28040 Madrid, Spain; eduardo.tolosana@upm.es

**Keywords:** shrub, milling, pelletisation, emissions, ash content, rockrose (*Cistus laurifolius* L.)

## Abstract

**Simple Summary:**

Rockrose (*Cistus laurifolius* L.) is a pyrophyte species with allelopathic effects that colonizes acid forest soils in the Mediterranean basin. Its continuous spread contributes to increasing wildfire risk, even more so in the current context of forecasts of increasingly intense and prolonged heat waves and droughts. Its mechanized collection and transformation into biofuels contributes to climate change mitigation, the economic development of rural areas and wildfire risk reduction. Biomass storage is an essential requirement in the supply chain of bio-refineries and biomass plants. This research aims to evaluate the influence of long-term outdoor storage (1 year) of baled rockrose shrub biomass on the quality of biofuels (30 mm milled material and pellets), on pre-treatment processes and on combustion emissions in an industrial boiler. After storage, no significant differences were observed in the biomass pre-treatment processes or in the emissions in an industrial combustion boiler. Biomass weight loss was 12% after storage. Some quality biofuels parameters improved, with the reduction in ash content being the most prominent aspect, which allowed 30 mm of milled material to be classified as I1, according to ISO 17225-9:2022, and rockrose pellets as class I3 for industrial use, according to the ISO 17225-2-2021 standard.

**Abstract:**

Biomass storage is an essential requirement in the supply chain of bio-refineries and power plants. This research aims to evaluate the influence of long-term outdoor storage (1 year) of baled rockrose (*Cistus laurifolius* L.) shrub biomass on biofuel’s quality, pre-treatment processes and on combustion emissions in an industrial boiler. The raw material was obtained from different rockrose shrublands in north central Spain. A total of 233 t_WM_ (tones of wet matter) of biomass were used to produce biofuels (30 mm of milled biomass and Ø 8 mm pellets) in the pre-treatment pilot plants at CEDER-CIEMAT. The combustion tests were conducted in an industrial moving grate boiler with a thermal power of 50 MWth, in a 17 MWe power plant. Outdoor storage improved some biofuel quality parameters, mainly the reduction in ash content, which allowed 30 mm of milled material to be classified as class I1 (ISO 17225-9:2022) and pellets as class I3 (ISO 17225-2-2021). No significant differences were observed in the total specific mass flow and energy consumption in the pre-treatment processes. The combustion tests had similar results, with the emissions being below the limits established in the directive (EU) 2015/2193. The results obtained indicated that the 1-year outdoor storage of rockrose-baled biomass, under Mediterranean conditions, was feasible for its subsequent use as biofuel.

## 1. Introduction

Forest biomass from woody shrub species is an important alternative feedstock for bioenergy in European countries [1,2]. Biofuels from shrub species can contribute to climate change mitigation, economic development of rural areas and wildfire risk reduction, especially in Mediterranean countries, such as Spain, where 8.4 million hectares (17% of the country) are occupied by shrublands. The sustainable management of shrub vegetation must be considered in rural development plans, both to achieve the succession of other plant formations with a higher evolutionary degree where appropriate, and to conserve these formations for their ecosystem value. Likewise, the orderly management of shrublands must be a priority in the fight against forest fires. Shrub clearing helps to eliminate the accumulation of woody fuels, favors the discontinuity of brush formations and establishes lines of defense in the surroundings of tree masses. Traditionally, mechanized clearing has been conducted with chain brush cutters, leaving the forest biomass on the ground. In Spain, there are shrub formations which have an anthropogenic origin, mainly derived from the abandonment of extensive livestock farming, which monospecifically occupy old grasslands, abandoned crop areas, and are susceptible, given their low slope, to mechanized harvesting.

In this country, energy production from biomass accounts for 28.7% of total primary energy production (5.5% in the electrical sector and 23.2% in the thermal sector) [3]. 

Many forest managers try to control wildfires by using prevention measures based on maintaining cleaned and cleared forests, grazing lands and paths, and promoting a reduction in fuel by shrubland clearing [4]. The uncontrolled spread of shrubs is often an environment conducive to the outbreak and proliferation of new fires in European Mediterranean countries [5,6,7,8]. In Spain, the number of fires sized over 1.0 hectare were 3760 per year (2010–2020 decade), affecting an annual total of 92,494 hectares, where 54,249 ha of those were covered by shrub [9].

Cistaceae bushes, one of the most abundant shrub formations in Spain, occupy more than 5.7 million ha, especially in non-calcareous soils, where they cover nearly 600,000 ha as the dominant species [10]. Rockrose (*Cistus laurifolius* L.), associated with oak, pine and holm oak forests in acidic or limestone soils, occupies 51,377 ha in the Iberian Peninsula as the dominant species and 680,856 ha as the secondary species [10,11]. This species abounds in the Iberian Peninsula (east, centre and south) and it is also distributed around the Aegean Sea, in Anatolia and the western Mediterranean (Tuscany, in Italy, Corsica, southern France and Morocco). Different mechanised harvesting trials on monospecific stands of rockrose (*Cistus laurifolius* L.) shrublands in central Spain [4,12] showed, in two years after clearing, an average of 83% reduction in fire spread speed, 63% reduction in heat per unit area, 83% reduction in fire line intensity and up to 77% reduction in flame length, which demonstrates the effectiveness of shrublands clearings [13].

Biomass is one of the few renewable energy sources that can be stored and can generate energy on demand [14]. Since the supply of forest material is not well aligned with consumption needs, storage is a necessary requirement in the supply chain, especially for bio-refineries and power plants [15]. In the case of shrublands, clearing is often hampered by factors related to weather, fauna and soil conditions, among others [4], so it is necessary to take maximum advantage of the periods when shrublands can be harvested, which generates a stock of raw material that needs to be stored.

During biomass storage, moisture content is considered the most important quality parameter, since it affects other factors such as the calorific value, storage properties and transport costs, and it is taken into account in the pricing of the fuel [16,17,18,19,20]. Natural drying is a cheap and easy method to reach an adequate moisture content in forest biomass [15,21]. Although there is a cost associated with immobilized capital, the developed heating markets consider differentiated prices of wood fuel, according to their moisture content [19], which can offset the opportunity cost associated with fixed capital. Natural drying can be carried out after harvesting and before milling to reduce the moisture content of biomass to less than 30% (wet basis), which makes it suitable for the supply of cogeneration or combined heat and power plants (>5 MW) and also for the supply of district heating systems and central heating systems (<5 MW) [20,22]. Moreover, the concentration of several inorganic elements, such as chlorine or sulphur, which are also biomass quality parameters that determine the classification of biomass according to ISO 17225-9:2022 [23] and ISO 17225-2:2021 [24] and can contribute to the corrosion of boilers and to emissions, can be modified during storage due to leaching by rainfall [25].

Some shrubs can be of an appropriate biomass to produce biofuels, such as pellets or milled material [8]. However, the composition of this kind of biomass could lead to fouling, slagging and high emissions of NOx or particulate matter, especially in domestic combustion devices, where systems for the mechanical cleaning or the abatement of particles and NOx do not exist or are very simple [8,26].

This research seeks to evaluate the influence of the 1-year outdoor storage of baled rockrose (*Cistus laurifolius* L.) shrub biomass on solid biofuel’s quality, on pre-treatment processes and on combustion emissions in an industrial boiler (50 MWth). The investigation was conducted in the framework of the Life+ Enerbioscrub project (Project LIFE13 ENV/ES/000660) which aimed to reduce the risk of forest fires by obtaining sustainable solid biofuels from shrublands of high flammability risk.

## 2. Materials and Methods

### 2.1. Raw Material

Rockrose (*Cistus laurifolius* L.) baled biomass from forest clearings in three shrublands located in Navalcaballo, Lubia and Torretartajo, Soria was used as raw material. The average standing biomass was 23.2 t_WM_/ha (tonnes of wet matter per hectare), which corresponds to 15.6 t_DM_/ha (tonnes of dry matter per hectare). 

A harvester–baler system (Biobaler WB55) was used to produce round bales (ϕ = 1.2 m, width = 1.2 m, density = 340 kg m^−3^) with an average productivity of 1.6 Mg_DM_ PMH^−1^ (Mg of dry matter per productive machine hour) and an average yield of 0.7 ha PMH^−1^ [12] (Figure A1). Harvesting yields ranged from 2.3 and 3.6 Mg_DM_/ha.

Baled biomass (233.27 Mg_WM_ [156.68 Mg_DM_]) was taken to CEDER-CIEMAT (Centre for the Development of Renewable Energy Sources) in Lubia, Soria for biomass storage study, pre-treatment tests and biofuel characterisation. Afterwards, 30 mm of milled biomass was used in combustion tests in an industrial power plant boiler in Garray, Soria, 30 km in distance from the cleared areas. The pre-treatment and combustion tests were repeated with just-harvested biomass and with biomass after 1-year of outdoor storage, following the same procedures.

### 2.2. Biomass Storage

Collected biomass was stacked outdoors at CEDER-CIEMAT in three-bale-high piles on a concrete deck with a perimeter fence. The baled biomass was stored for one year from 3/2/2016 to 31/1/2017. Prolonging outdoor storage more than one year might not be recommended since bale strings tend to break, and consequently, picking up loose bales with broken strings could be a risk during de-stacking work.

The storage area has a continental Mediterranean climate [27,28], with an average annual temperature of 10.7 °C and an average annual rainfall of 472 mm over the last ten years (2009–2017). Meteorological data were obtained from Lubia Weather Station, placed 50 m in distance from the storage area.

Rockrose baled biomass was stored for 1 year to evaluate the influence of outdoor storage on biomass weight, dry matter loss, and biofuel’s physicochemical properties.

Biomass dry matter variation, expressed as a percentage, was calculated as the difference between dry biomass weight before and after the storage period, divided by the mass before the storage. For this, 45 rockrose bales, stored outdoors in a 3-bale piles (Figure A2) were weighed individually before and after storage using a weighing hook (500 kg ± 200 g) attached to the bucket arm of a tractor.

Solid biofuels (30 mm milled biomass and Ø 8 mm pellets) were produced with just-harvested biomass and with 1-year-stored biomass following the same pre-treatment processes. Biofuel samples for subsequent analytical characterisation were obtained according to the following procedure.

(a) 30 mm of milled biomass: Three bales of just-harvested biomass were randomly selected for subsequent shredding to 30 mm. A combined sample of milled material, consisting of 6 sub-samples of 2 kg each, was prepared for laboratory analysis. After the storage period, three bales, stacked at different heights, were randomly selected to be shredded and sampled for analysis. In addition, to evaluate the bale-moisture-content variation throughout the storage period, one bale was shredded to 30 mm every 2 months, from those placed in the central height of the piles, and a biomass sample was taken following the protocol described above. Moisture content was measured using an oven-drying method at 105 °C according to ISO 18134-2. A combined sample of milled material, consisting of 3 sub-samples of 1 kg each, was prepared for moisture-content analysis.

(b) Ø 8 pellets: Three pelletisation tests were carried out with just-harvested biomass and another three tests with 1-year-stored biomass. In each test, 6 subsamples of 2 kg each were taken from moving material at the outlet of the bagging bin of the pelletisation pilot plant described in Section 2.3. Afterwards, sub-samples were mixed, and a combined sample was taken for analytical characterisation. The same sampling protocol was used to sample stored biomass pellets.

### 2.3. Biofuels Production

As mentioned in the previous section, two different biofuels were produced from rockrose bales: 30 mm of milled biomass to feed an industrial boiler and Ø 8 mm pellets to evaluate their quality and possible application in the domestic or industrial sector. The pre-treatment tests were performed with just-harvested biomass and with biomass stored outdoors for one year, following the same procedures.

For the production of biofuels, the following biomass pre-treatment plants were used. It should be noted that, since the bales’ initial moisture content was 38.5%, a previous drying process in a rotary dryer was needed before just-harvested biomass shredding.

(a)Communition equipment. A 90 kW pre-shredder was used to reduce rockrose bales in size in order to ensure a proper boiler feed and optimum efficiency of the combustion system. In this equipment, the material was pressed against a monorotor via a hydraulic feeder with a pusher stroke of 1100 mm. This rotor (ϕ 450 mm, 1400 mm length) was provided with 102 embedded blades (40 × 40 mm) that milled the biomass by passing it through a 30 mm mesh. The biofuels produced were used to power an industrial combustion boiler without previous sieving and drying processes (Figure A3-left).

Subsequently, part of the shredded biomass was milled for further pelletisation. A 75 kW hammer mill with a 4 mm mesh was used to mill shredded bales for pellet production (Figure A3-center).

(b)Pelletisation plant

Pelletisation tests were carried out in a pilot plant with a 30 kW flat die press, Ø 500 mm die diameter and 4.4 compression ratio flat die (ratio of the effective working length of the die holes [35 mm] to the die hole diameter [8 mm]) (Figure A3-right).

The following amount of biomass was used before storage and the same quantity was also processed after storage:Pre-shredder tests: Three batches of 80 bales (35 t_WM_ per lot) were crushed to 30 mm for the industrial boiler. Part of the obtained material (4.3%) was post-grinded for pelletisation tests.Post-grinder tests: Three batches of 30 mm shredded biomass (1500 kg_WM_ per batch) were processed in the hammer mill to 4 mm for pelletisation tests.Pelletisation tests: Three batches of 4 mm biomass (1500 kg_WM_ per batch) were pelletised to obtain Ø 8 mm pellets.

During the biomass pre-treatment tests (shredding, milling and pelletisation), the specific mass flow and the specific energy consumed in the different processes were recorded when the equipment reached steady state conditions. The specific mass flow (MF, kg_DM_ h^−1^ kW^−1^) was obtained by dividing the mass of the processed material (kg_DM_) by the time used to shred, mill or pelletise it (h) and by the power of the corresponding equipment (kW). To obtain the specific energy (E, kWh Mg_DM_^−1^), the active electric energy (kWh) required by the shredder, mill and pelletisation press was divided by the mass of the processed material (Mg_DM_) in each case.

### 2.4. Biomass Combustion

The combustion tests were conducted in an industrial moving grate boiler with a thermal power of 50 MW_th_, in a 17 MWe power plant. The boiler used allows a steam production of 58.5 Mg h^−1^ overheated at 93 bar and 487 °C, which generates an electricity production of 108,333 MWh per year. Designed to be fueled by forest biomass, mainly wood logs, pruning waste bales and milled wood, it was fed for the trial with milled rockrose. The combustion gases were filtered through a multicyclone and bag filter to treat a gas flow of 74,000 Nm^3^ h^−1^ before being emitted to the atmosphere.

Combustion tests were performed with 30 mm of milled material during a steady state period of 4 h, following the same procedure for just-harvested biomass and biomass after 1-year outdoor storage, in order to compare emissions (NOx, SO_2_ and particles). During the combustion tests, different settings were modified with the aim of adjusting the installation of operational parameters to the rockrose fuel, and emissions were measured under steady state conditions. Previously, the combustion of a forest fuel commonly used in the power plant was also monitored during the same time and under steady state conditions. Continuous measurement of the gaseous composition of exhaust gases was carried out with a portable Fourier Transform Infrared (FTIR) Spectroscopy analyser. Furthermore, three discontinuous measurements of total solid particles (TSP) were performed, applying EN 13284-1:2017 [29], through isokinetic samplings.

### 2.5. Analytical Procedures

Biofuels were analysed in the Laboratory of Biomass Characterisation at CEDER-CIEMAT. Analytical samples were prepared according to the ISO 14780:2017; Sample preparation standard: Geneva, Switzerland, 2017, by means of homogenisation, division, grinding and drying. In order to evaluate changes in physical and chemical properties, the following parameters were analysed: moisture content, calorific value, ultimate analysis (C, H, N, S and Cl), major and minor elements, ash content and ash composition, all by following the standards and analytical methods shown in Table 1.

## 3. Results

### 3.1. Biomass Storage

After 1-year outdoor storage with climate conditions shown in Table 2, rockrose bale piles remained stable, bales kept most of their leaves, but their density decreased from 340 kg m^3^ to 222 kg m^3^ and most of the baling ropes were loose and some of them had broken (Figure A4). Biomass moisture content decreased from 38.5% to 22.5% after the first 4 months of storage (February–May 2016) and from 22.5% to 17.8% after the next 8 months (June 2016–January 2017) (Table 3). The biomass dry matter variation after 1-year of outdoor storage was 12.3% (Table 3).

### 3.2. Biofuels Production

The values of the specific mass flow (MF) and specific energy (E) of the pre-treatment tests (shredding, milling and pelletisation processes) for just-harvested biomass and biomass after 1-year outdoor storage are shown in Table 4.

#### 3.2.1. Communition Tests

Shredding results at 30 mm indicated that there were not significant differences before and after the storage period. Relative increases of 3.6% in specific mass flow (MF), from 16.6 to 17.2 kg h^−1^ kW^−1^ (d.b.) and of 5.1% in specific energy (E), from 11.7 to 12.3 kWh Mg^−1^ (d.b.), were observed when using the stored biomass.

Milling results at 4 mm were also similar in terms of specific mass flow (10.9 vs. 9.0 kg h^−1^ kW^−1^) and specific energy demand before and after storage (56.4 vs. 54.7 kWh Mg^−1^ [d.b.]), with similar values of biomass moisture content. An analysis of variance (ANOVA) of MF and E in shredding and milling processes was carried out, taking into account the storage duration factor. Significance levels (α) of the F-tests are shown in Table 5. A significance level of less than 0.05 indicates that a factor has a statistically significant effect on the measured property at 95.0% probability. No differences in shredding and milling were found between just-harvested biomass and biomass after 1-year outdoor storage at 95% probability (α = 0.05), although 4 mm milling MF would show differences after 1 year of outdoor storage if the statistical contrast was carried out at 90% probability (α = 0.10).

#### 3.2.2. Pelletisation Tests

Similar results of specific mass flow were obtained when pelletising just-harvested biomass and biomass after 1 year of outdoor storage (6.4 vs. 6.3 kg h^−1^ kW^−1^ [d.b.]) (Table 5). The specific energy was slightly higher (6.1% in relative percentage) with stored biomass (127.4 vs. 135.2 kWh Mg^−1^ [d.b.]), although no significant differences were found at 95% probability (α = 0.05).

The total specific mass flow and energy consumption in the shredding, milling and pelletisation processes were similar before and after storage (33.9 vs. 32.6 kg h^−1^ kW^−1^ [d.b.] and 195.5 vs. 202.2 kWh Mg^−1^ [d.b.]), taking into account that an initial drying in a rotary dryer was needed before just-harvested biomass shredding, with an energy consumption of 21 kWh Mg^−1^ (d.b.).

### 3.3. Biofuels Characterisation

The physical and chemical characterisation of the rockrose biofuels (30 mm of milled material and Ø 8 mm pellets) with just-harvested biomass and biomass after 1-year outdoor storage can be seen in Table 5.

The cumulative particle size distribution of the 30 mm milled rockrose before and after storage was similar (Figure 1). The most notable difference was observed in the fraction of fine particles (<3.5 mm) of milled rockrose before and after storage (10.5% vs. 5.5%). Calorific values in the dry basis remained almost constant after storage in both biofuels. Ash content decreased from 4.7 to 2.8% in the case of 30 mm milled rockrose and from 4.2 to 3.0% in the case of pellets. Cl content was not modified during storage and S content decreased. Trace elements also decreased after storage in both biofuels.

As can be seen in Table 5, the characteristic melting temperatures of the solid biofuels studied were quite similar, with the deformation temperature (DT) being higher than 1200 °C. The DT marks the beginning of melting and is therefore frequently used as the temperature of reference in laboratories and thermochemical facilities. Therefore, no problems of sintering were expected due to the fact that the most of combustors work at temperatures below 1200 °C. This good behavior of rockrose could be due to a high proportion of CaO, especially in the milled rockrose after storage, which generates the highest DT, higher than 1450 °C.

### 3.4. Biomass Combustion

The industrial boiler (50 MW_th_) fed with 30 mm milled rockrose was operated without mechanical problems, allowing a homogeneous boiler-feeding and stable combustion conditions. Problems associated with the melting of ashes were not observed, which is consistent with the ash fusibility temperatures (Table 5). Gaseous and particle emissions during the steady state period of the combustion tests, referring to dry basis and reference O_2_ of 6 v.%, are shown in Table 6.

## 4. Discussion

### 4.1. Biomass Storage

The decrease in moisture content was greater in the first 4 months, with a cumulative rainfall of 259.6 mm, than in the remaining 8 months, with 172.1 mm rainfall, mainly concentrated in November (Figure A5). As can be seen in Table 2, weather in 2016 was similar to the average during the period 2009–2017, with 14.5% more precipitation (540 mm) and the same annual mean temperature (10.7 °C). This proves that outdoor-baled rockrose storage, in a Mediterranean climate, is an effective solution to dry biomass that requires little capital investment. In contrast, in the North of the Iberian Peninsula, with an Atlantic climate, 1-year pine woodchip storage is not a good solution to dry biomass, since the moisture content of the pile increases with storage time [46].

Regarding biomass dry matter variation after 1-year outdoor storage (12.3%), similar or greater losses were referred to in previous studies of Mediterranean shrub open storage trials along a 1-year period (14% with baled broom or 14–30% with mulched gorse) [4].

After storage, biomass ash content was reduced by 40.4%. Ash chemical composition indicated that the content of Si, Al, Fe, P and Ti, elements typically present in the soil, were reduced after storage, which could explain that the decrease in ash content was due to the action of the wind and the biomass-washing process by rainfall, together with the handling during de-stacking. The content of C and H did not vary after storage, indicating that there was no biological degradation of the stored biomass. This was confirmed by the biomass net calorific value, which did not change after storage (Table 5). 

According to baled rockrose biomass total net costs, including harvesting and baling with Biobaler, bale gathering, loading and transport at destination (76.76 € Mg_DM_^−1^) [12], the biomass dry matter variation (12.3%) involved an increment of 13.4% in the cost of a dry biomass tone at destination.

Other outdoor storage systems that minimise biomass losses should be tested in the future. It would be interesting to assess biomass losses by placing round bales in rows of one height, with the flat side of one bale next to another. The rows should face south-southwest to allow for maximum exposure to the sun. The distance between the rows should be at least one meter to allow the wind to pass through.

On the other hand, it would also be interesting to estimate biomass losses by placing a waterproof and breathable tarpaulin over the prismatic bale stacks.

### 4.2. Biofuels Production

After analysing the specific mass flow and specific energy results of the biomass pre-treatment processes performed in triplicate before and after storage, no significant differences were observed in any of the shredding, milling and pelletisation tests.

Figures of specific mass flow and specific energy of shredding, milling and pelletisation processes, before and after the storage, were of the same order as the reported values to obtain 30 mm of shredded material and Ø 8 mm pellets from broom bales in the same equipment [4,8]. As in previous works with other shrub biomass, small variations in communition results could be related to the differences in the milled biomass moisture content and with the lack of homogeneity of the baled material [8].

### 4.3. Biofuels Characterisation

Biomass storage contributed to improving some biofuel quality parameters, with the reduction in ash content and trace elements being the most notable aspects (Table 5). In the case of 30 mm milled material, the ash content reduction after storage (≤3%), together with the decrease in Ni content (from 18.0 to 5.0 mg kg^−1^ d.b., below I1 class limit ≤ 10 mg kg^−1^ d.b.), allowed for the classification of this biofuel as class I1, according to ISO 17225-9:2022, dedicated to the specification of graded hog fuel and wood chips for industrial use (Table 7). The rest of the parameters were widely fulfilled. It should be noted that the stored biomass was shredded with new blades in the pre-shredder rotor, which were less alloyed in Cr and Ni than the previous ones. Therefore, the higher content of these elements in the just-harvested biomass could be due to the contamination caused by the blades’ wear. On the other hand, the fraction of fine particles of milled rockrose (graded hog fuel) was particularly high (10.5% < 3.5 mm) (Table 6 and Figure 1) and a prior screening process before its use would be advisable to reduce fine particles.

According to the ISO 17225-2-2021 standard, the pellets obtained after storage were suitable for industrial use (I3 class) due to the high ash content. Since this property is the only limiting factor for the rockrose pellets to be used in the domestic sector, with an appropriate pre-treatment, e.g., fine particles separation, which involves a reduction in the ash content of the pellets [47,48,49], they could be classified in category B class (≤2% ash content) of the aforementioned standard, for use in commercial and residential applications that demand high-quality products with minimum ash content (Table 7). Without prior screening, the use of rockrose biofuels seems to be more suitable in industrial boilers, where cleaning systems for abatement, such as multicyclones and fabric filters, are usually present. 

### 4.4. Biomass Combustion

Rockrose combustion in the industrial boiler fulfilled the limits established in the standard directive (EU) 2015/2193 of the European Parliament and of the council of 25 November 2015 on the limitation of emissions of certain pollutants into the air from medium combustion plants (Table 7). In addition, no differences in combustion emissions were observed between biofuels from just-harvested rockrose and rockrose after 1-year outdoor storage, with emissions being similar to those of forest fuels commonly used in the biomass plant. With the use of a bag filter, the TSP (total suspended particles) emitted through the stack were similar between milled rockrose before and after storage, despite having a higher ash content than the milled rockrose from just-harvested biomass.

Furthermore, compared to other studies, NOx emissions obtained from milled rockrose (367 and 338 mg/Nm^3^ d.b. at 6 v.% O_2_ as reference, Table 6) were lower than that registered using shrub biomass from broom (457 mg/Nm^3^ d.b. at 6 v.% O_2_, [8]), but were higher than that measured during the combustion of milled pine (172 mg/Nm^3^ d.b. at 6 v.% O_2_, [8]), which is in concordance with their nitrogen composition (Table 5). SO_2_ emissions from rockrose combustion (13 and 11 mg/Nm^3^ d.b. at 6 v.% O_2_, Table 6) are in the same range as that obtained when combusting milled broom (8.5 mg/Nm^3^ d.b. at 6 v.% O_2_; 6.2 mg/Nm^3^ d.b. at 10 v.% O_2_, [8]) and milled pine (29 mg/Nm^3^ at 6 v.% O_2_; 21 mg/Nm^3^ at 10 v.% O_2_, [8]), as their sulphur composition is similar (Table 5).

## 5. Conclusions

The results obtained in this study indicate that a long-term outdoor store (1 year) of rockrose-baled biomass in north-central Spain, under Mediterranean continental climate, is feasible for its subsequent use as biofuel.

Outdoor storage on a concrete deck proved to be an effective solution for biomass drying, assuming 12% biomass weight loss. It improved some biofuel quality parameters, with a reduction in ash content being the most notable aspect. This allowed the classification of 30 mm of milled material as class I1, according to ISO 17225-9:2022, dedicated to the specification of graded hog fuel and wood chips for industrial use, and rockrose pellets obtained as class I3 for industrial use, according to the ISO 17225-2-2021 standard.

No significant differences were observed in the total specific mass flow and energy consumption in the rockrose shredding, milling and pelletisation processes before and after storage. The combustion of the rockrose biomass before and after storage had similar results and did not produce higher emissions than one of the fuels commonly used in the industrial boiler, being therefore below the limits established in the directive (EU) 2015/2193.

In forthcoming research, inert element separation systems implementing fine particle screening after shredding and milling during pre-treatment processes will be tested to reduce the ash content of the resulting pellets and shredded material, in order to bring them closer to the commercial quality of non-industrial pellets.

## Figures and Tables

**Figure 1 biology-12-01451-f001:**
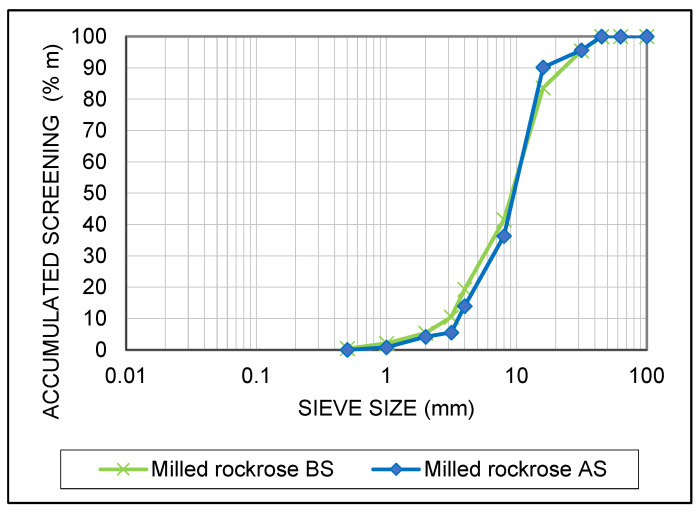
Cumulative particle size distribution of 30 mm milled rockrose before storage (BS) and after 1-year outdoor storage (AS).

**Table 1 biology-12-01451-t001:** Standards and analytical methods used at CEDER-CIEMAT Biomass Characterisation Laboratory.

Property	Technique of Analysis	Standard
Sampling	Solid biofuels sampling	ISO 21945 [30]
Analytical sample preparation	Homogenisation, division, drying and grinding	ISO 14780 [31]
Bulk density	Mass of a known volume	ISO 17828 [32]
Mechanical durability	Drum rotation	ISO 17831-1 [33]
Fines content	Manual sieving (3.15 mm mesh)	ISO 18846 [34]
Particle size distribution	Separation into defined size fractions (sieves)	ISO 17827-1 [35] ISO 17827-2 [36]
Moisture	Drying at 105 °C	ISO 18134-2 [37]
Ash	Calcination at 550 °C	ISO 18122 [38]
Volatile matter	Heating at 900 °C	ISO 18123 [39]
C, H, N S and Cl	Elemental analysis (TruSpec, Leco) Ion chromatography (883 Basic IC Plus, Methrom) after sample combustion and lixiviation	ISO 16948 [40] ISO 16994 [41]
Calorific value	Automatic calorimeter (C5003, Ika Verke)	ISO 18125 [42]
Major elements	Microwave digestion (Ethos Pro, Milestone) and ICP-AES (Jarrell ash, Thermo Scientific, Madrid, Spain)	ISO 16967 [43]
Minor elements	Microwave digestion (Ethos Pro, Milestone) and ICP-MS (iCAP Q, Thermo Scientific) Thermal decomposition/gold amalgamation/absorption spectrophotometry (DMA-80, Milestone) for Hg	ISO 16968 [44]
Ash melting behaviour	Optical heating microscopy (Hesse instruments, Osterode am Harz, Germany)	ISO 21404 [45]

**Table 2 biology-12-01451-t002:** Atmospheric mean values during the storage period, year 2016, and 2009–2017 period.

Parameter	Unit	Storage Period	2016	2009–2017
Mean temperature	°C	11.0	10.7	10.7
Absolute maximum temperature	°C	35.6	35.6	34.7
Absolute minimum temperature	°C	−11.7	−7.5	−10.2
Mean maximum temperature	°C	17.2	17.3	17.5
Mean minimum temperature	°C	4.6	4.8	4.3
Precipitation	mm	431.7	540.2	471.8
Direct solar radiation	MWh m^−2^	1.42	1.40	1.4
Relative humidity	%	68.4	69.5	66.8
Wind speed	m s^−1^	1.9	2.0	n.a.
Wind direction	Degrees	261.4	258.9	n.a.

n.a.: not available.

**Table 3 biology-12-01451-t003:** Moisture content and dry matter variation during 1-year outdoor storage period (3 February 2016 to 31 January 2017).

Description	Value
Initial moisture content (%)	38.5
2-month moisture content (%)	30.3
4-month moisture content (%)	22.5
6-month moisture content (%)	21.4
8-month moisture content (%)	19.1
10-month moisture content (%)	20.6
Final moisture content (%)	17.8
Initial stored wet biomass (kg_WM_)	20,748
Initial stored dry biomass (kg_DM_)	12,760
Initial wet weight per bale (kg_WM_ per bale)	461.1
Initial dry weight per bale (kg_DM_ per bale)	283.5
Final stored wet biomass (kg_WM_)	13,611
Final stored dry biomass (kg_DM_)	11,188
Final wet weight per bale (kg_WM_ per bale)	302.5
Final dry weight per bale (kg_DM_ per bale)	248.6
Dry matter variation (kg_DM_)	1572
Dry matter variation (%)	12.3

**Table 4 biology-12-01451-t004:** Results of rockrose communition and pelletisation tests with just-harvested biomass and biomass after a 1-year outdoor storage period.

Pre-Treatment Process	Test 1	Test 2	Test 3	A	SD	%RSD	α
Shredder (30 mm)							
Just-harvested biomass							
MF (kg h^−1^ kW^−1^) (d.b.)	17.8	19.3	12.8	16.6	3.4	20.5	
E (kWh Mg^−1^) (d.b.)	11.7	10.6	12.9	11.7	1.2	9.8	
M (wt.%) (w.b.)	25.0	30.0	23.0	26.0	3.6	13.9	
Biomass after 1-year storage							
MF (kg h^−1^ kW^−1^) (d.b.)	19.3	17.4	15.0	17.2	2.2	12.5	
E (kWh Mg^−1^) (d.b.)	11.7	11.7	13.4	12.3	1.0	8.0	
M (wt.%) (w.b.)	17.5	18.8	18.3	18.2	0.7	3.6	
*p*-value of the F-test (95% prob.)							
MF (kg h^−1^ kW^−1^) (d.b.)							0.809
E (kWh Mg^−1^) (d.b.)							0.574
Post-grinder (4 mm)							
Just-harvested biomass							
MF (kg h^−1^ kW^−1^) (d.b.)	10.6	9.7	12.3	10.9	1.3	12.2	
E (kWh Mg^−1^) (d.b.)	58.1	57.3	53.8	56.4	2.3	4.1	
M (wt.%) (w.b.)	22.1	13.8	13.5	16.5	4.9	29.6	
Biomass after 1-year storage							
MF (kg h^−1^ kW^−1^) (d.b.)	9.1	9.0	8.9	9.0	0.1	1.0	
E (kWh Mg^−1^) (d.b.)	58.5	53.5	52.0	54.7	3.4	6.2	
M (wt.%) (w.b.)	19.2	14.3	16.6	17.9	1.8	10.3	
*p*-value of the F-test (95% prob.)							0.071
MF (kg h^−1^ kW^−1^) (d.b.)							0.505
E (kWh Mg^−1^) (d.b.)							
Pelletisation (Ø 8 mm)							
Just-harvested biomass							
MF (kg h^−1^ kW^−1^) (d.b.)	6.0	5.8	7.3	6.4	0.8	12.8	
E (kWh Mg^−1^) (d.b.)	128.0	137.2	117.1	127.4	10.1	7.9	
M (wt.%) (w.b.)	10.0	9.1	11.0	10.0	1.0	9.5	
Biomass after 1-year storage							
MF (kg h^−1^ kW^−1^) (d.b.)	6.2	7.3	5.5	6.3	0.9	14.3	
E (kWh Mg^−1^) (d.b.)	131.0	117.0	157.7	135.2	20.7	15.3	
M (wt.%) (w.b.)	11.5	11.0	10.2	10.9	0.7	6.0	
*p*-value of the F-test (95% prob.)							
MF (kg h^−1^ kW^−1^) (d.b.)							0.965
E (kWh Mg^−1^) (d.b.)							0.588

A: Average value; MF: specific mass flow; E: specific energy; M: moisture content (wt.%: weight %; d.b.: dry basis; w.b.: wet basis; SD: standard deviation; RSD: relative standard deviation (%); *p*-value: significance level (α) of the F-tests pre-treatment processes (95% of probability).

**Table 5 biology-12-01451-t005:** Physical and chemical characterisation of 30 mm milled rockrose and Ø 8 mm rockrose pellets, before and after storage.

Parameter	Unit	Milled Rockrose		Rockrose Pellets	
		BS	AS	BS	AS
Diameter	mm	-	-	8	8
Moisture content	wt.% (w.b.)	16.5	10.9	7.2	9.4
Bulk density	kg m^−3^ (w.b.)	280	250	700	670
Mechanical durability	%	-	-	97.3	99.2
Fines content	%	10.5	5.5	0.2	0.3
Calorific values:					
GCV	MJ kg^−1^ (d.b.)	19.61	19.85	19.9	19.9
GCV	MJ kg^−1^ (w.b.)	16.38	17.69	18.5	18.0
NCV	MJ kg^−1^ (d.b.)	18.33	18.55	18.9	18.6
NCV	MJ kg^−1^ (w.b.)	14.9	16.26	17.4	16.6
Inmediate analysis:					
Ash	wt.% (d.b.)	4.7	2.8	4.2	3.0
Volatile matter	wt.% (d.b.)	78.8	80.4	78.6	79.9
Ultimate analysis:					
Carbon	wt.% (d.b.)	49.3	49.6	50.2	50.4
Hydrogen	wt.% (d.b.)	5.9	6.0	5.8	5.9
Nitrogen	wt.% (d.b.)	0.45	0.42	0.40	0.40
Sulphur	wt.% (d.b.)	0.04	0.03	0.04	0.02
Chlorine	wt.% (d.b.)	0.01	0.01	0.01	0.01
Ash composition:					
Al_2_O_3_	wt.% (d.b.)	2.2	1.5	2.9	1.7
BaO	wt.% (d.b.)	0.13	0.19	0.16	0.15
CaO	wt.% (d.b.)	18	42	23	28
Fe_2_O_3_	wt.% (d.b.)	2.0	0.62	1.6	1.0
K_2_O	wt.% (d.b.)	4.4	6.0	6.7	6.0
MgO	wt.% (d.b.)	1.8	3.4	2.7	2.8
Mn_2_O_3_	wt.% (d.b.)	0.52	0.77	0.64	0.61
Na_2_O	wt.% (d.b.)	0.14	0.20	0.21	0.15
P_2_O_5_	wt.% (d.b.)	2.1	1.8	2.8	2.4
SO_3_	wt.% (d.b.)	1.8	1.6	1.5	1.4
SiO_2_	wt.% (d.b.)	51	19	35	31
SrO	wt.% (d.b.)	0.031	0.059	0.039	0.047
TiO_2_	wt.% (d.b.)	0.18	0.088	0.23	0.12
ZnO	wt.% (d.b.)	0.12	0.11	0.072	0.074
Sum	wt.% (d.b.)	84.0	80.3	76.8	76.2
Trace elements on biomass					
As	mg kg^−1^ (d.b.)	0.21	<0.10	0.19	<0.10
Cd	mg kg^−1^ (d.b.)	0.27	0.28	0.29	0.25
Cr	mg kg^−1^ (d.b.)	8.1	1.6	3.9	1.8
Cu	mg kg^−1^ (d.b.)	5.6	3.1	5.0	2.5
Pb	mg kg^−1^ (d.b.)	1.6	1.7	1.7	1.7
Hg	mg kg^−1^ (d.b.)	0.005	0.004	0.005	0.004
Ni	mg kg^−1^ (d.b.)	18.0	5.0	11.0	2.9
Zn	mg kg^−1^ (d.b.)	26	22	22	17
Ash melting behavior					
SST	°C	1220	1390	1200	1210
DT	°C	1230	>1450	1210	1230
HT	°C	1250	>1450	1250	1260
FT	°C	1280	>1450	1260	1260

BS: before storage; AS: after storage; GCV: gross calorific value; NCV: net calorific value; wt.%: weight %; w.b.: wet basis; d.b.: dry basis. SST: shrinkage staring temperature; DT: deformation temperature; HT: hemisphere temperature; FT: flow temperature.

**Table 6 biology-12-01451-t006:** Emissions during the steady state of the combustion tests in an industrial boiler (50 MW_th_) referring to dry gas basis and reference O_2_ of 6%v.

Biofuel	NO_x_ (mg/Nm^3^)		SO_2_ (mg/Nm^3^)		TSP (mg/Nm^3^)	
	A	SD	A	SD	A	SD
Commonly used fuel	367	23	11	1.7	13	4.2
Milled rockrose (BS)	367	23	13	2.1	17	5.5
Milled rockrose (AS)	338	22	11	2.0	16	5.8
Directive (EU) 2015/2193 and RD 1042/2017 limits	650		200		30	

%v.: % volume; BS: before storage; AS: after storage; A: average value; SD: standard deviation; NO_x_: NO + NO_2_, shown as NO_2_; TSP: total solid particles.

**Table 7 biology-12-01451-t007:** Physical and chemical characterisation of 30 mm milled rockrose and Ø 8 mm rockrose pellets after biomass storage and limits established by ISO 17225-2 and ISO 17225-9.

Parameter	Unit	Milled Rockrose	ISO 17225-9	Rockrose Pellets	ISO 17225-2	
		AS	I1 Class	AS	B Class	I3 Class
Diameter	mm	-	-	8	8	8
Moisture content	wt.% (w.b.)	10.9	≤45	9.4	≤10	≤10
Bulk density	kg m^−3^ (w.b.)	250	-	670	≥600	≥600
Mech. durability	%	-	-	99.2	≥96.5	≥96.5
Fine particles	%	5.5		0.3	≤1.0	≤6.0
Calorific value:						
NCV	MJ kg^−1^ (w.b.)	16.3		16.6	≥16.5	≥16.5
Inmediate analysis:						
Ash	wt.% (d.b.)	2.8	≤3	3	≤2	≤3
Ultimate analysis:						
Nitrogen	wt.% (d.b.)	0.42	≤0.5	0.4	≤1.0	≤0.6
Sulphur	wt.% (d.b.)	0.03	≤0.05	0.02	≤0.05	≤0.05
Chlorine	wt.% (d.b.)	0.01	≤0.05	0.01	≤0.03	≤0.1
Trace elements on biomass						
As	mg kg^−1^ (d.b.)	<0.10	≤1	<0.10	≤1	≤2
Cd	mg kg^−1^ (d.b.)	0.28	≤2.0	0.25	≤0.5	≤1.0
Cr	mg kg^−1^ (d.b.)	1.6	≤20	1.8	≤10	≤15
Cu	mg kg^−1^ (d.b.)	3.1	≤30	2.5	≤10	≤20
Pb	mg kg^−1^ (d.b.)	1.7	≤10	1.7	≤10	≤10
Hg	mg kg^−1^ (d.b.)	0.004	≤0.1	0.004	≤0.1	≤0.1
Ni	mg kg^−1^ (d.b.)	5	≤10	2.9	≤10	≤10
Zn	mg kg^−1^ (d.b.)	22	≤100	17	≤100	≤100
Ash fusibility						
DT	°C				≥1100	

AS: biofuels made with 1-year stored biomass; NCV: net calorific value; wt.%: weight %; w.b.: wet basis; d.b.: dry basis. DT: deformation temperature.

## Data Availability

Data supporting reported results can be found at CEDER-CIEMAT.

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
