# Peer review of "The Influence of the Long-Term Outdoor Storage of Rockrose (Cistus laurifolius L.) Shrub Biomass on Biofuel’s Quality, Pre-Treatment and Combustion Processes"

_biology, 2023, doi:10.3390/biology12111451_

Round 1
Reviewer 1 Report (Previous Reviewer 3)
Comments and Suggestions for Authors
The manuscript is quite interesting and relevant, however I have few comments to improve it further:
As authors mentioned that 17% of Spain is under this shrubland, did authors take into account the ecological implications if this biomass resource is used for biofuel production. Please explain that.
This poses similar challenge as in case of crop residues, there is abundance but collection is difficult due to widespread, this shrub also grows with others in wild thus harvesting and collection would be challenging and can up the cost
Can authors include more information about biomass yield in terms of ha?
In conclusion, authors could propose that what can be done in future to improve the effectiveness of this pretreatment, which subsequently could improve biomass quality.
Comments on the Quality of English Language
It can be sent to a native English speaker
Author Response
Thank you very much for taking the time to review this manuscript. Please find the detailed responses below and the corresponding corrections highlighted in track changes in the re-submitted files.
- The introduction has been completed with the requested information related with the ecological implications of shrub collection for biofuels production.
- All the cited references are relevant to the research. Research related to the mechanized use of scrubland for biomass is a very new aspect in Mediterranean countries. Experiences in Spain started in the framework of the LIFE+ENERBIOSCRUB project (LIFE13 ENV/ES/000660; http://enerbioscrub.ciemat.es/) and the bibliographic citations, both of mechanized harvesting and biomass pretreatment and combustion derive from this project, so it is necessary to include them in this research article.
- The methodology has been completed with the requested information related with biomass collection yield.
- In the discussion section, a comparison of the combustion of rockrose biomass with respect to other similar shrubs and other biomasses has been included.
- A native English professor has reviewed the research paper.

Reviewer 2 Report (Previous Reviewer 4)
Comments and Suggestions for Authors
Abstract
Line 80: “store” needs to be storage.
Materials and methods
Line 291-292: Not clear how the moisture measurement was done. Was it using oven drying method? If yes, what was the number of samples for each bale and how long was it dried for at what temperature?
Line 304-305: Check language for clarity. “1-year outdoor storage biomass” is not grammatically correct.
Results
Table 4: Check the placement of p-values in Table 4.
Comments on the Quality of English Language
The quality of English needs to be improved as there are many instances where grammar is not right.
Author Response
Thank you very much for taking the time to review this manuscript. Please find the detailed responses below and the corresponding corrections highlighted in track changes in the re-submitted files.
- The methodology has been completed with the requested information related with the moisture measurement.
- The language and tables have been checked for clarify.
- A native English professor has reviewed the research paper.

Reviewer 3 Report (New Reviewer)
Comments and Suggestions for Authors
please see the attached file

Comments on the Quality of English Language
only minor spell checking
Author Response
Thank you very much for taking the time to review this manuscript and considering it is a well-designed study with promising results. Please find the detailed responses below and the corresponding revisions/corrections highlighted/in track changes in the re-submitted files.
- The introduction has been completed with information related with the ecological implications of shrub collection for biofuels production.
- The methodology has been completed with information requested by reviewer 1 related with biomass collection yield.
- In the discussion section, a comparison of the combustion of rockrose biomass with respect to other similar shrubs and other biomasses has been included.
- All the figures in the paper have been replaced by others with better resolution.
- A native English professor has reviewed the research paper.

Reviewer 4 Report (New Reviewer)
Comments and Suggestions for Authors
Manuscript in which the authors carry out a study on how the storage of rockrose for one year affects its application as a solid biofuel, mainly characterisation and combustion. The work presented is unique and novel, given the scarcity of studies on this raw material. It is remarkable the methodology and equipment used, which is the result of the extensive experience that this research group has in its field of study. It should be noted that the authors include a specific discussion section in the manuscript, which should be revised in order to be considered an appropriate discussion section. In addition, a number of minor errors of various nature have been detected during the present review, which should be corrected (see specific comments to the authors).
According to the above mentioned I give different considerations for the authors in the following section "Specific comments to the authors" to complete and specify the information and shown.
"Specific comments to the authors":
Keywords: include rockrose and/or Cistus rifolius
Ref [3] and [4]: are not sources of information considered sufficiently verified to be used as a reference. Replace with references that have been published after a peer review process (mainly articles or conferences) or information published by governmental institutions (e.g. MAPAMA).
Ref [5]: when you click on the link, status not found. Revise.
Line 159: [12,11]. Correct order [11,12].
Line 176-180: include references to corroborate this statement.
Table 1 and Table 2: Replace º(underlined) by ° (revise in document complete)
Section 2.5. The authors detail the methodology followed in the characterisation of rockrose as a solid biofuel. An important property is the fusibility of the ash and the results are not shown. This phenomenon can cause fouling and slagging when this type of biomass is used in combustion applications. This is a well-known problem, as explained by the authors in the introduction. Furthermore, ash fusibility is listed as an informative parameter in the ISO 17225-2 standard, which is used by the authors later as a guide to classify the solid biofuel obtained, but as this parameter is not shown, the characterisation is incomplete.
Table 5: the authors show durability and fines results but standards and analytical methods of this test are not included in Table 1.
Lines 470-481: the main results obtained from the different characterisations carried out are highlighted. The explanations given by the authors in lines 475, 476, 477 and 479, which justify the results obtained should be removed from this section and included in the corresponding discussion part (4.1, lines 516-522).
Lines 500-501: This explanation may be adequate but it is not a statement that should be in the discussion of results section of this study. What is said is not obtained as a consequence of the study shown in this manuscript. It is a recommendation that supports the fact that the study was carried out in a time lapse of 1 year. Therefore these lines should be moved from this section and included in section 2.2.
Section 4.3: in this section the authors have to include a table showing the results obtained by the present study and the limits established by ISO 17225-2 and ISO 17225-9.
Section 4.4: this section is incomplete. A comparison of the combustion of this biomass with respect to other similar shrubs and other biomasses in general obtained from previous work should be shown and discussed.
Author Response
Thank you very much for taking the time to review this manuscript. Please find the detailed responses below and the corresponding corrections highlighted in track changes in the re-submitted files.
- The introduction has been completed with the requested information related with the ecological implications of shrub collection for biofuels production.
- All the cited references are relevant to the research. Research related to the mechanized use of scrubland for biomass is a very new aspect in Mediterranean countries. Experiences in Spain started in the framework of the LIFE+ENERBIOSCRUB project (LIFE13 ENV/ES/000660; http://enerbioscrub.ciemat.es/) and the bibliographic citations, both of mechanized harvesting and biomass pretreatment and combustion derive from this project, so it is necessary to include them in this research article.
- The methodology has been completed with the requested information related with biomass collection yields.
- In the discussion section, a comparison of the combustion of rockrose biomass with respect to other similar shrubs and other biomasses has been included.
- All the figures in the paper have been replaced by others with better resolution.
- A native English professor has reviewed the research paper.

Round 2
Reviewer 4 Report (New Reviewer)
Comments and Suggestions for Authors The authors have correctly modified the manuscript to incorporate the suggestions made.
This manuscript is a resubmission of an earlier submission. The following is a list of the peer review reports and author responses from that submission.
Round 1
Reviewer 1 Report
Comments and Suggestions for Authors
Explain how biofuel is produced in this work. Currently the biofuel production method is not clear.
Any comparison study using different shrub plant?
Table 7: SD vs std dev. Are both the same?
Author Response
Response to Reviewer 1 Comments
Point 1: Explain how biofuel is produced in this work. Currently the biofuel production method is not clear.
Response 1: Two different biofuels were produced from rockrose bales: 30 mm milled biomass and Ø 8 mm pellets.
- The 30 mm milled biomass was produced in a 90 kW pre-shredder that milled the biomass by passing it through a 30 mm mesh. This equipment is described in section 3.2.1 and shown in the suplementary Figure S2 (left).
- The Ø 8 mm pellets were produced in a pelletisation pilot plant with a 30 kW flat die press, Ø 500 mm die diameter and 4.4 compression ratio flat die (ratio of the effective working length of the die holes [35 mm] to the die hole diameter [8 mm]). This equipment is described in section 2.3.2 and shown in Figure S2 (right).
The methodology section of the paper has been reorganised and some heading have been changed in order to clarify biofuels production processes.
Point 2: Any comparison study using different shrub plant?
Response 2:
Given the scarcity of reference studies on the energy use of the Mediterranean scrub in Spain, the publications cited are those derived from the LIFE+ ENERBIOSCRUB project http://enerbioscrub.ciemat.es/:
- Mediavilla, I.; Borjabad, E.; Fernández, M.J.; Ramos, R.; Pérez, P.; Bados, R.; Carrasco, J.E.; Esteban, L.S. Biofuels from broom clearings: Production and combustion in commercial boilers. Energy 2017, 141, 1845-1856. https://doi.org/10.1016/j.energy.2017.11.112
- Esteban, L.S.; Bados, R.; Mediavilla, I. Sustainable management of shrub formations for energy purposes. 2019. Available online: http://enerbioscrub.ciemat.es/documents/210922/222403/Manual+Gesti%C3%B3n+Arbustos/9c8c61bc-2792-497c-a0d3-2433d0b84ac7/ (accessed on 26 June 2023).
Point 3: Table 7: SD vs std dev. Are both the same?
Response 3: That's right. SD is the same as Std. dev. This last abbreviation has been removed from Table 7 footnotes.

Reviewer 2 Report
Comments and Suggestions for Authors
In this manuscript biomass storage from rockrose was evaluated under environmental conditions. The authors stored the biomass for one year in field condition and the determined their potential as raw material for biofuel production before and after the storage. I have serious doubts about the real contribution of this manuscript.
1. The methodology is confused for me. Sometimes explain the procedure from Pile B after for Pile A. They described first the installation, and in my opinion this data must be given together with the procedure. I recommend to re-write this section according to the analysis made and to indicate what biomass was employed in each case. In the Biomass pre-treatment section, the authors indicated that the same quantity was employed before and after storage: “three lots of 80 bales were crushed” (line 172-174), but later in the text I understood something different. Pile B was used for biofuel properties and that pile had only 45 bales (lines 119-121)… So, what is the correct idea?
2. I would like to know how the authors determined that Pile A (with 220 bales) is adequate. I checked the reference ([25] Dooley et al., 2018) and that data can’t be obtained from this reference. Moreover, the authors only give us the dimension of Pile B (with 45 bales; Figure S3), but I feel that the dimensions of pile A should also have been considered in the manuscript, especially because the authors indicated that biomass to measure the moisture content (every two months) were taken from this pile, and always from the central height (lines 199-200). On the other hand, why they only used one bale each two months (6 in total) if they had 220 bales? I understood that pile A was only used to measure moisture content, and pile B to evaluate biofuel properties. Why didn't they make the piles the same?
3. Why the authors determined to use ANOVA for statistical analysis if they only had two samples? Why they only made the analysis for palletization? In this sense, the data from table 5 corresponded to each replicate (test 1, test 2 and test 3)? Why did not make statistical analysis for chemical and physical parameters and another characteristic?
4. The storage period was between February/2016 and January/2017 and the precipitation in this period was (172.1 mm), while during year 2016 the precipitation was 540.2… So, the precipitation in January/2016 was 368.1 mm, is this correct? (Table 2)
5. The discussion is weak. They only gave their results again without bibliographic comparation. Moreover, one of the few comparisons made, with woodchips ([28] Casal et al., 2010), would be biased because that is a very different biomass, with different dimensions too.
6. Finally, the fact of the blades were contaminated (line 359-360) make me doubt about the results from before storage biomass.
Author Response
Response to Reviewer 2 Comments
In this manuscript biomass storage from rockrose was evaluated under environmental conditions. The authors stored the biomass for one year in field condition and determined their potential as raw material for biofuel production before and after the storage. I have serious doubts about the real contribution of this manuscript.
- The contribution of this article is relevant in Mediterranean countries such as Spain, where there is a high risk of forest fires and prevention work includes mechanized clearing of the scrub, leaving the biomass on the ground without further use. The collection of this biomass and its subsequent storage make it possible to have a resource that can be used as biofuel
- On the other hand, in Europe there is a need to sustainably mobilize new biomass resources. In the last 20 years the use of bioenergy has tripled in the EU (from 41 Mtoe in 2000 to 117 Mtoe in 2020), providing renewable energy for heating (74%) and also for transport and electricity (26%) (Bioenergy Europe, 2021). This entails the need to search for new sources of biomass. The sustainable use of the scrub can be an incentive to improve local forest management, helping to reduce the risk and intensity of forest fires (Esteban et al., 2017; Mediavilla et al., 2017). According to the Spanish Strategy for the Development of Forest Biomass (MARM, 2010), the forest biomass potential in Spain is close to 6.6 million tMS/year (tons of dry matter per year), of which 4.5 million tMS/year correspond to wooded forest and 2.1 million tMS/year to scrub.
- In Mediterranean countries it is not allowed to carry out mechanised forestry work during the season of high fire risk (July, August and September). This implies the need to collect biomass, store it during the summer for drying, and use it on demand the rest of the year. Currently, there are no studies that evaluate the outdoor storage of scrub biomass for a long period of time. For the bioenergy industrial sector, it is important to evaluate the influence of outdoor rockrose storage on the energy consumption of the biofuels production processes, on the biofuels quality and on their behavior in combustion boilers for industrial use.
Point 1: The methodology is confused for me. Sometimes explain the procedure from Pile B after for Pile A. They described first the installation, and in my opinion this data must be given together with the procedure. I recommend to re-write this section according to the analysis made and to indicate what biomass was employed in each case. In the Biomass pre-treatment section, the authors indicated that the same quantity was employed before and after storage: “three lots of 80 bales were crushed” (line 172-174), but later in the text I understood something different. Pile B was used for biofuel properties and that pile had only 45 bales (lines 119-121)… So, what is the correct idea?
Response 1:
Thank you for your recommedations. The methodology section has been modified according to your criteria and is now clearer and more precise.
Regarding the amount of material processed to produce biofuels, the same amount of biomass was used before and after storage (35 tWM). In section 2.5.1, where it said that 265 bales of stored biomass were used for biofuels, it should read 240 bales. This is corrected in the new version.
Finally, it has not been considered relevant to distinguish between piles A and B, as both piles were in the same place and the disposition of the bales was similar. Pile B (45 bales) was separated from Pile A (220 bales) to weigh each bale individually before and after storage. Bales from Pile A were taken periodically to analyse biomass moisture content in order to keep Pile B intact, as all its bales were identified with their corresponding noted weight. After storage, the biomass sample for analysis was taken from Pile B, although it could equally well have been taken during the shredding of the bales from Pile A. The final version of the document does not differentiate between piles A and B.
Point 2: I would like to know how the authors determined that Pile A (with 220 bales) is adequate. I checked the reference ([25] Dooley et al., 2018) and that data can’t be obtained from this reference. Moreover, the authors only give us the dimension of Pile B (with 45 bales; Figure S3), but I feel that the dimensions of pile A should also have been considered in the manuscript, especially because the authors indicated that biomass to measure the moisture content (every two months) were taken from this pile, and always from the central height (lines 199-200). On the other hand, why they only used one bale each two months (6 in total) if they had 220 bales? I understood that pile A was only used to measure moisture content, and pile B to evaluate biofuel properties. Why didn't they make the piles the same?
Response 2:
Thank you for pointing this out. Having checked the bibliography of Doodley et al, 2018, I have checked that the citation is indeed incorrect. The sentence and the reference have been omited.
Pile A (220 bales) was 32x3.6x3.6 m. This pile dimensions were not considered in the manuscript because its bales were not weighted individually. As I have already indicated in the previous point, in the new version of the paper the two piles of bales will not be differentiated.
To measure the moisture content of the biomass, it was considered sufficient to shred one bale every two months, as the average weight of just harvested bale was 461 kgMH and after one year of storage, 249 kgMH. A combined sample of 30 mm milled material, consisting of 6 sub-samples of 2 kg each was prepared for laboratory analysis.
As previously mentioned, the 45 bales in Pile B were separated from the rest of the bales for individual measurement and control.
Point 3: Why the authors determined to use ANOVA for statistical analysis if they only had two samples? Why they only made the analysis for palletization? In this sense, the data from table 5 corresponded to each replicate (test 1, test 2 and test 3)? Why did not make statistical analysis for chemical and physical parameters and another characteristic?
Response 3:
Analysis of variance (ANOVA) of specific mass flow and specific energy in shredding, milling and pelletisation processes were carried out since three tests of each process were carried out before and after storage. However, statistical analyses of the physico-chemical properties were not carried out because two biomass samples were analysed, one at the beginning and one at the end of storage.
Point 4: The storage period was between February/2016 and January/2017 and the precipitation in this period was (172.1 mm), while during year 2016 the precipitation was 540.2… So, the precipitation in January/2016 was 368.1 mm, is this correct? (Table 2)
Response 4:
Thank you for your comment. When I checked the data in Table 2, I tested out that the meteorological information for the storage period was wrong. In January 2016 the rainfall was 120.6 mm.
The data in Table 2 has been changed and now reads as follows:
Table 2. Atmospheric mean values during the storage period, year 2016 and 2009-2017 period.
Parameter |
Unit |
Storage period |
2016 |
2009-2017 |
Mean temperature |
ºC |
11.0 |
10.7 |
10.7 |
Absolute maximum temperature |
ºC |
35.6 |
35.6 |
34.7 |
Absolute minimum temperature |
ºC |
-11.7 |
-7.5 |
-10.2 |
Mean maximum temperature |
ºC |
17.2 |
17.3 |
17.5 |
Mean minimum temperature |
ºC |
4.6 |
4.8 |
4.3 |
Precipitation |
mm |
431.7 |
540.2 |
471.8 |
Direct solar radiation |
MWh m-2 |
1.42 |
1.40 |
1.4 |
Relative humidity |
% |
68.4 |
69.5 |
66.8 |
Wind speed |
m s-1 |
1.9 |
2.0 |
n.a. |
Wind direction |
Degrees |
261.4 |
258.9 |
n.a. |
n.a.: not available.
Point 5: The discussion is weak. They only gave their results again without bibliographic comparation. Moreover, one of the few comparisons made, with woodchips ([28] Casal et al., 2010), would be biased because that is a very different biomass, with different dimensions too.
Response 5:
Is correct. The discussion lacks bibliographical references because there are currently no studies on the storage of bales of Mediterranean scrub. This type of forest vegetation is generally not collected and left on the ground after mechanised clearing.
Point 6: Finally, the fact of the blades were contaminated (line 359-360) make me doubt about the results from before storage biomass.
Response 6:
In any grinding process, the biomass is contaminated with the cutting elements of the mill. The grinding involves several stages, not only the grinding stage in the pilot plant (commercial grinding) but also the grinding in the laboratory to generate an analytical sample with a particle size of less than 1 mm.
In this work, the results of chromium and nickel, especially the latter, were elevated in the biomass before storage. As a consequence, new blades were purchased for the pilot plant mill and a new laboratory mill was purchased with the ability to replace the steel rotor and its steal sieve with a titanium rotor and titanium sieve.
Consequently, the sentence of section 4.3: “It should be noted that the stored biomass was shredded with new blades in the pre-shredder rotor, which were less alloyed in Cr and Ni than the previous ones. Therefore, the higher content of these elements in the just-harvested biomass could be due to the contamination caused by the blade´s wear” will be replaced by:
“In order to try to reduce the high content of Ni and Cr in the stored biomass, typically elements derived from steel, the blades were changed in the pre-shredder rotor and a new mill with a titanium rotor and sieve (1 mm sieve hole) was purchased in the laboratory. Therefore, the higher content of these elements in the just-harvested biomass could be due to the contamination caused by the different grinding processes.”

Reviewer 3 Report
Comments and Suggestions for Authors
The study is quite interesting and authors have described well by covering all the relevant aspects. However, there are few suggestions which can help to improve the manuscript further.
Overall authors need to restructure introduction, there is hardly anything about biomass quality, authors need to link the role of storage in terms of biomass quality and its not only about moisture and calorific value, there are other inorganic constituents of biomass which are influenced due to storage such content of Cl. These constituents of biomass are highly important in terms of biomass quality, please include these aspects, identify the research gap, why this storage method is relevant here.
- L48-49 Prevention measures instead of policies
- L51 I don't think uncontrolled concentration is right expression, would be better to say something like uncontrolled spread
- L54, is it scrub or shrub?
In discussion part, authors should reflect on short comings of the study and what can be done in future
Comments on the Quality of English Language
English language needs serious improvements
Author Response
Response to Reviewer 3 Comments
The study is quite interesting and authors have described well by covering all the relevant aspects. However, there are few suggestions which can help to improve the manuscript further.
Point 1: Overall authors need to restructure introduction, there is hardly anything about biomass quality, authors need to link the role of storage in terms of biomass quality and its not only about moisture and calorific value, there are other inorganic constituents of biomass which are influenced due to storage such content of Cl. These constituents of biomass are highly important in terms of biomass quality, please include these aspects, identify the research gap, why this storage method is relevant here.
Response 1: Please provide your response for Point 1.
The introduction has been restructured and the role of storage in terms of biomass quality has been incorporated.
Point 2: L48-49 Prevention measures instead of policies.
Response 2:
Thank you for the suggestion. The change has been done.
Point 3: L51, I don't think uncontrolled concentration is right expression, would be better to say something like uncontrolled spread
Response 3:
Thank you for the suggestion. The expression has been changed.
Point 4: L54, is it scrub or shrub?
Response 4:
Scrub is correct. Shrubland is open vegetation where shrubs are the tallest plants. Scrub is denser, with a continuous or near-continuous canopy of shrubs and small trees.
Point 5: In discussion part, authors should reflect on short comings of the study and what can be done in future.
Response 5:
Thank you for the proposal.
Other outdoor storage systems that minimise biomass losses should be tested in the future. It would be interesting to assess biomass losses by placing round bales in rows of one height, with the flat side of one bale next to the next. The rows should face south-southwest to allow maximum exposure to the sun. The distance between the rows should be at least one metre to allow the wind to pass through.
On the other hand, it would also be interesting to estimate biomass losses by placing a waterproof and breathable tarpaulin over the prismatic bale stacks.
Point 6: English language needs serious improvements.
Response 6: The English style is pending review by the Gabinete Lingüístico de la Universidad Complutense de Madrid.

Reviewer 4 Report
Comments and Suggestions for Authors
General comments:
Significance of the biomass could be discussed more with how it is being utilized currently. The volume of this biomass production could support the significance. The flow of information is also not in order making it hard for the reader as information from one section is referenced in another and so on. Reorganization of the paper is necessary especially in the methods section. Results need to be discussed with more scientific reasons than is currently done.
Specific comments:
Introduction:
Line 48-51: The sentence is long making it confusing. It can be broken down for clarity.
Line 54: Check for typo.
Line 65: Change “what” to “which”.
Materials and methods: What was the bulk density of the bales produced? As the bulk density is an important factor that affects the storability of biomass bales.
Line 115: Justification for the suitable pile size not clear.
Line 119: It is unclear why the two pile sizes were so drastically different from each other.
Line 122: Instead of defining the “climate” as “cold” and “hot”, providing temperature ranges for the winter and summer months would provide more clarity.
Line 125: Check spelling.
Line 136-137: Incomplete sentence.
Results:
Table 3: The dry matter variation is calculated for the whole pile rather than per bale. Its more appropriate to calculate it per bale as these are discrete units and not a pile of chopped biomass.
Table 4: It would be more appropriate to report the composition of the biomass in percentage for before and after storage. The current table is difficult to understand and doesn’t make much sense.
Line 261-286: Lacks the discussion for the trend seen for the different mass flows and specific energy consumption for the three processes for the pre- vs post-storage biomass.
Table 7: Lack of combustion tests on the pellets formed makes it seem incomplete.
Conclusions:
Line 387: It is difficult to say that this is a cheap method for storage without accounting for the economic loss due to loss of biomass material.
Comments on the Quality of English Language
The grammar and language clarity of the paper is poor making it difficult to understand in many instances.
Author Response
Response to Reviewer 4 Comments
Point 1: General comments:
Significance of the biomass could be discussed more with how it is being utilized currently. The volume of this biomass production could support the significance. The flow of information is also not in order making it hard for the reader as information from one section is referenced in another and so on. Reorganization of the paper is necessary especially in the methods section. Results need to be discussed with more scientific reasons than is currently done.
In the introduction, the importance of forest biomass as a biofuel for the production of electricity and heat has been highlighted. The most relevant physico-chemical properties for the use of biofuels in the domestic and industrial sector have also been described. The methodology section has been reorganised.
The discussion section on biofuels lacks bibliographical references to compare the results obtained with other tests carried out with scrubland biofuels, but unfortunately, studies on the use of Mediterranean scrubland for biomass are scarce and most of them have been carried out within the ENERBIOSCRUB project "Sustainable management of shrub formations for energy use" (http://enerbioscrub.ciemat.es/), the project in which this paper is framed.
Point 2: Introduction:
Response 2:
Line 48-51: The sentence is long making it confusing. It can be broken down for clarity.
Thank your for your indication. The sentence has been broken down for clarity.
“Many forest managers try to control wildfires by prevention policies based on maintaining cleaned and cleared forests, grazing lands and paths [3] and also promote the reduction of fuel by shrublands clearing [4,5]. The uncontrolled concentration of shrub is often an environment conducive to the outbreak and spread of new fires in the Europe-an Mediterranean countries [6-9].”
Line 54: Check for typo.
Thanks for your review. There was a verb was missing in the sentece. It has been corrected as follow:
“In Spain, the number of fires sized over 1.0 hectare were 3,760 per year (2010-2020 decade), affecting an annual total of 92,494 hectares, 54,249 ha out of them were covered by scrub [10].”
Line 65: Change “what” to “which”.
The word has been changed. Thank you.
Different mechanised harvesting trials on monospecific stands of rockrose (Cistus laurifolius L.) shrublands in central Spain [5,13] showed, in two years after clearing, an average of 83% reduction in fire spread speed, 63% reduction in heat per unit area, 83% reduction in fire line intensity and up to 77% reduction in flame length, which demonstrates the effectiveness of shrublands clearings [14].
Point 3: Materials and methods:
Response 3:
What was the bulk density of the bales produced? As the bulk density is an important factor that affects the storability of biomass bales.
Bales bulk density has been included in this sentence:
Round bales (Ï•=1.2 m, width=1.2 m, density=340 kg m3) were made by a harvester-baler system called Biobaler WB55 (Figure S1).
Line 115: Justification for the suitable pile size not clear.
The bales were stacked three high as this was the maximum height that could be reached with the tractor used to store the bales. The depth of the piles (3 bales, 3.6 m) was considered adequate to favor aeration between bales, what avoids self-combustion and anaerobic fermentation. The length of the pile was determined by the size of the available concrete deck.
Line 119: It is unclear why the two pile sizes were so drastically different from each other.
Piles A and B were built to the same height and depth. The only difference is that Pile B was made shorter (7.2m vs. 32m) because it was the fraction of bales that were individually weighed before and after storage.
In the new paper version it was not considered relevant to distinguish between piles A and B, as both piles were in the same place and the disposition of the bales was similar. Pile B (45 bales) was separated from Pile A (220 bales) to weigh each bale individually before and after storage. Bales from Pile A were taken periodically to analyse biomass moisture content in order to keep Pile B intact, as all its bales were identified with their corresponding noted weight in order to evaluate biomass losses after one year outdoor storage.
The stored biomass sample for analysis was taken from Pile B, although it could equally well have been taken during the milling of the bales from Pile A. The final version of the document does not differentiate between piles A and B as it is not considered relevant.
Line 122: Instead of defining the “climate” as “cold” and “hot”, providing temperature ranges for the winter and summer months would provide more clarity.
The sentence has been changed to this other:
The storage area has a continental Mediterranean climate [26,27], with an average annual temperature of 10.7 oC and an average annual rainfall of 472 mm over the previous ten years (2009-2017) (Table 2).
Line 125: Check spelling.
The title “Biomass pre-treatment instalations” is correct. However, Materials and Methods setion has been rewritten and this title has been eliminated.
Line 136-137: Incomplete sentence.
There was a mistake in the sentece because "it was used" was left over. The sentence is like this:
“Subsequently, part of the shredded biomass was milled for further pelletisation. A 75 kW hammer mill with a 4 mm mesh was used to mill shredded bales for pellet production (Figure S2-center).”
Point 4: Results
Response 4:
Table 3: The dry matter variation is calculated for the whole pile rather than per bale. Its more appropriate to calculate it per bale as these are discrete units and not a pile of chopped biomass.
Dry matter variation was calculated for the whole pile (pile B). The 45 bales of pile B were individually weighed before and after storage without shredding. Biomass dry matter variation, expressed as a percentage, was calculated as the difference between the dry biomass weight before and after storage, divided by the dry mass before storage. Samples of 30 mm milled rockrose, shredded before and after storage were used for moisture content analysis to express the percentage in dry matter.
Table 4: It would be more appropriate to report the composition of the biomass in percentage for before and after storage. The current table is difficult to understand and doesn’t make much sense.
Thank you very much. We have considered your suggestion and believe that you are correct. As Table 4 is based on Table 6, we have removed Table 4 from the article and the comments on the differences in the physico-chemical composition of the biomass have been made based on the characterisation of the 30 mm shredded material in Table 6.
Line 261-286: Lacks the discussion for the trend seen for the different mass flows and specific energy consumption for the three processes for the pre- vs post-storage biomass.
Thank you very much. An initial paragraph in the section 4.2 Biofuels production has been added. It indicates that there were no significant differences in the three biomass pre-treatment processes:
“After analysing the specific mass flow and specific energy results of the biomass pre-treatment processes performed in triplicate before and after storage, no significant differences were observed in any of the shredding, milling and pelletisation tests”.
Table 7: Lack of combustion tests on the pellets formed makes it seem incomplete.
In the framework of the ENERBIOSCRUB project, which is the subject of this article, pellet combustion tests were carried out in a 40 kW commercial moving grate boiler. The results have not been included in this article because they were already published in the following reference:
Borjabad, E., Mediavilla, I., Pascual, A., García, S., Fernández, M.J., Carrasco, J.E., Esteban, L.S., Ramos, R. Influence of outdoor storage of shrub biomass on emissions and slagging during its combustion. 25th European Biomass Conference and Exhibition, 12-15 June 2017, Stockholm, Sweden.
Point 5: Conclusions
Response 5:
Line 387: It is difficult to say that this is a cheap method for storage without accounting for the economic loss due to loss of biomass material.
This sentence was intended to indicate that outdoor biomass storage is a cheap system that does not require major investments in infrastructure, such as agricultural straw sheds. Nor has any kind of tarpaulin been used to cover the bales. Only a concrete floor has been used to avoid possible contamination from the ground. In addition, the concrete prevents waterlogging areas by favouring the drainage of rainwater.
The sentence has been modified in the new version of the article as follows:
“Outdoor storage on a concrete deck proved to be an effective and cheap solution for biomass drying, assuming 12% biomass weight loss.”
According to baled rockrose biomass total net costs, including harvesting and baling with Biobaler, bale gathering, loading and transport at destination (76.76 € MgDM-1) [13], the biomass dry matter variation (12.3%) involved an increment of 13.4% in the cost of a dry biomass tone at destination.
The grammar and language clarity of the paper is poor making it difficult to understand in many instances.
The introduction and methodology sections of the article have been improved to make it easier to read and understand. The English style is pending review by the Gabinete Lingüístico de la Universidad Complutense de Madrid.

Round 2
Reviewer 3 Report
Comments and Suggestions for Authors
Authors have revised it sufficiently, after minor English check, manuscript can be accepted
Comments on the Quality of English Language
Minor English check is required